# Progress in the Application of CNN-Based Image Classification and Recognition in Whole Crop Growth Cycles

**Feng Yu** [1,2,†], **Qian Zhang** [2,†], **Jun Xiao** [1,*], **Yuntao Ma** [2], **Ming Wang** [2], **Rupeng Luan** [2], **Xin Liu** [2], **Yang Ping** [2], **Ying Nie** [2], **Zhenyu Tao** [2] and **Hui Zhang** [2]

- [1] School of Artificial Intelligence, University of Chinese Academy of Sciences, Beijing 100049, China; yufeng21@mails.ucas.ac.cn
- [2] Institute of Data Science and Agricultural Economics, Beijing Academy of Agriculture and Forestry Sciences, Beijing 100097, China; zhangq@agri.ac.cn (Q.Z.); yuntao.ma@cau.edu.cn (Y.M.); niey@agri.ac.cn (Y.N.)
- [*] Correspondence: xiaojun@ucas.ac.cn
- [†] These authors contributed equally to this work.

**Abstract:** The categorization and identification of agricultural imagery constitute the fundamental requisites of contemporary farming practices. Among the various methods employed for image classification and recognition, the convolutional neural network (CNN) stands out as the most extensively utilized and swiftly advancing machine learning technique. Its immense potential for advancing precision agriculture cannot be understated. By comprehensively reviewing the progress made in CNN applications throughout the entire crop growth cycle, this study aims to provide an updated account of these endeavors spanning the years 2020 to 2023. During the seed stage, classification networks are employed to effectively categorize and screen seeds. In the vegetative stage, image classification and recognition play a prominent role, with a diverse range of CNN models being applied, each with its own specific focus. In the reproductive stage, CNN's application primarily centers around target detection for mechanized harvesting purposes. As for the post-harvest stage, CNN assumes a pivotal role in the screening and grading of harvested products. Ultimately, through a comprehensive analysis of the prevailing research landscape, this study presents the characteristics and trends of current investigations, while outlining the future developmental trajectory of CNN in crop identification and classification.

**Keywords:** convolution neural network; crop classification; crop recognition; whole crop growth cycle

## 1. Introduction

The precise recognition and classification of crops form the bedrock of agricultural intelligence and automation, benefiting from the advancements in image recognition and machine learning technologies. Effective seed identification and classification play a pivotal role in crop breeding endeavors. By accurately identifying the growth status of plants, one can monitor the overall crop development and facilitate precise fertilization practices. Additionally, the identification and enumeration of plants and fruits hold significant potential for the application of automated harvesting systems, such as autonomous picking robots.

The process of image classification and recognition throughout the entirety of the crop growth cycle presents certain challenges. Firstly, the same crop exhibits noticeable morphological variations across different stages of growth. Furthermore, distinct crops may exhibit similar visual characteristics during the seedling stage, such as cabbage and cauliflower, cucumber, and pumpkin. Secondly, diverse production settings, including fields, facilities, and orchards, present intricate backgrounds that often encompass elements such as soil, stones, weeds, and various structures. Thirdly, the quality of the captured images may be adversely affected by factors such as illumination, shooting angles, and weather conditions. Lastly, in the context of large-scale cultivation, branches, leaves, flowers, fruits, and canopies can intertwine and obstruct each other, posing challenges

in accurately delineating the complete boundaries of plants, flowers, and fruits within the images.

Conventional image-processing methods traditionally emphasize shallow image features, including morphology, gray difference, and color [1]. However, these methods are constrained by artificial features and fail to fully exploit the rich potential of image characteristics. In contrast, the convolutional neural network (CNN) possesses the capability to independently learn image features and uncover latent potential. By leveraging extensive training data, CNN continually enhances its proficiency in extracting deep image features. As a result of its remarkable recognition accuracy and robustness, CNN has emerged as the dominant network in the realm of crop image classification and recognition. Within the realm of agricultural image processing, the convolutional neural network (CNN) stands as the prevailing framework among deep learning models. This can be attributed to its remarkable capacity to rapidly and precisely extract highly discriminative representations from diverse image features. However, it is important to acknowledge that CNN's effectiveness hinges upon the availability of ample datasets accompanied by annotated labels. This necessitates an arduous process of collecting extensive datasets and undertaking manual labeling efforts.

Existing scholarly investigations have primarily focused on two main areas within the domain of crop phenotyping: the utilization of Multiscale-Deep-Learning [2,3] and the advancements in agriculture IoT technologies [4]. Moreover, specific tasks such as unmanned aerial vehicle (UAV) applications [5], crop yield measurement [6], and weed identification [7] have been the subject of comprehensive analysis. Nevertheless, it is crucial to recognize the interconnectedness of different stages throughout the crop growth cycle. The research methodologies employed in one stage may hold valuable insights and applicability to other stages. Thus, the objective of this study is to consolidate and summarize the advancements made in the application of convolutional neural networks (CNN) across the entire growth cycle of crops during the past three years. To ensure the timeliness and relevance of our findings, a comprehensive search was conducted in the Web of Science core library using pertinent keywords in January 2023. This yielded a total of 1056 pertinent studies reported between 2020 and 2023. From this dataset, a meticulous manual screening process was undertaken, resulting in the inclusion of 213 papers that met the criteria for our study.

This study initially introduces the concept of the entire crop growth cycle, the basic framework of CNN, and representative algorithms pertaining to different branches. Subsequently, the progress of CNN application is analyzed and compared across the entire crop growth cycle, while common issues and special cases encountered in CNN application are also addressed. Furthermore, the application characteristics of CNN in various image-processing tasks are thoroughly analyzed and compared. Lastly, the challenges and prospects of CNN in crop identification and classification are assessed.

## 2. Whole Crop Growth Cycle and CNN

### 2.1. Whole Crop Growth Cycle

This study encompasses an examination of various types of crops, including grain, cash, feed, and medicinal crops. The term "whole crop growth cycle" denotes the comprehensive progression from the initial sowing stage to the acquisition of new crop seeds. Typically, this cycle is divided into three distinct stages based on their defining characteristics. Additionally, due to the necessity of grading and the identification of commercialized crop products in the postharvest stage, this particular stage has been incorporated into the framework. The definition of each stage within the entire crop growth cycle is as follows:

- Seed stage: This stage encompasses the period from the fertilization of maternal egg cells to the germination of seeds. During this stage, crops undergo embryonic development and seed dormancy.
- Vegetative stage: from seed germination to the differentiation of flower buds.
- Reproductive stage: Following a series of changes during the vegetative stage, crops initiate the development of flower buds on the growth cone of their stems. Subsequently, the crops blossom, bear fruit, and eventually form seeds.
- Postharvest stage: This stage involves the harvesting of mature crop plants, seeds, fruits, roots, and stems. Once harvested, these crops undergo screening and grading to facilitate subsequent sale or seed production.

### 2.2. CNN and Its Development

The prototype of CNN traces its origins back to the LeNet-5 network model, initially proposed by Le Cun et al. [8] in 1998. A conventional CNN architecture (Figure 1) comprises several layers, namely, input, convolution, activation, pooling, and fully connected layers [9]. In this study, the CNN utilized represents a series of algorithm models developed within this framework, based on these foundational principles.

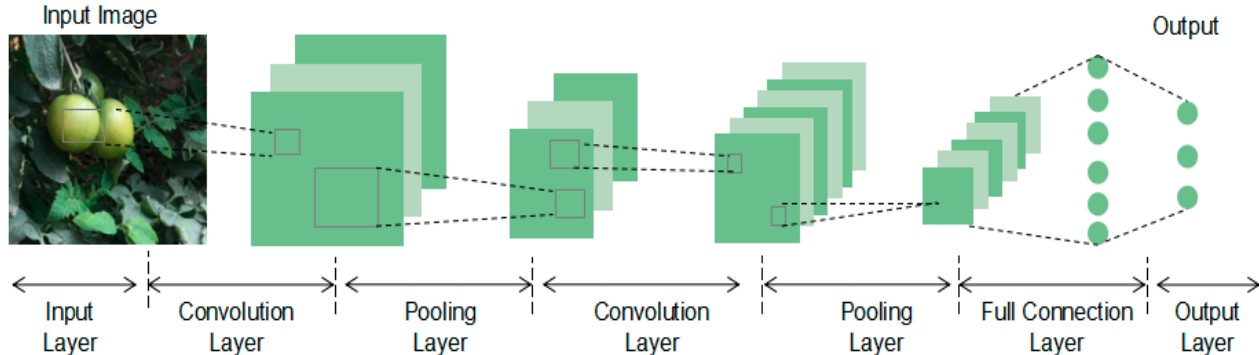

**Figure 1.** Basic Framework of CNN. The input layer reads images as input to the CNN network; the convolution layer achieves feature enhancement; the activation layer performs nonlinear transformations on image features; the pooling layer performs down-sampling to sparse the feature map to reduce computation; the full connection layer reduces feature information loss through re-fitting; the output layer returns image categories.

The introduction of AlexNet [10] in 2012 signified the emergence of deep learning and its subsequent advancements. Subsequently, algorithms such as GoogLeNet, VGG, ResNet, and others, built upon the CNN framework, attained significant success in the ImageNet visual recognition challenge. Consequently, CNN has evolved into the predominant network framework within the realm of deep learning, especially in the field of computer vision. Figure 2 depicts the progression of CNN development and the exploration of representative algorithms. Based on their functionalities, CNN can be categorized into three distinct groups:

- The Classification network solely determines the category to which the entire picture belongs, without providing object positions or object count calculations.
- The Target-detection network precisely identifies the category and location of a specific object within the image. It can be categorized into one-stage and two-stage algorithms, with the one-stage algorithm being faster and the two-stage algorithm being more accurate.
- The Segmentation network classifies and segments all the pixels in the image at a pixel level. It can be further categorized into semantic and instance segmentation.

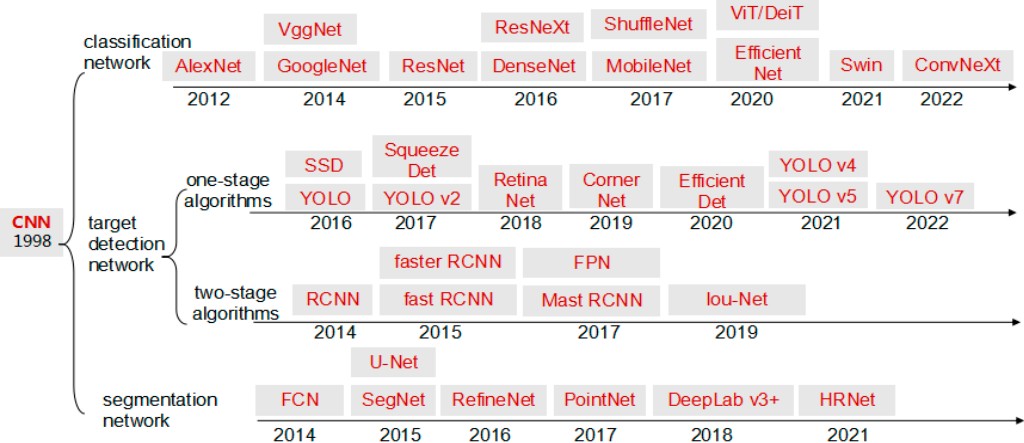

**Figure 2.** Development and representative algorithm of CNN.

## 3. Progress of CNN Applications in Crop Growth Cycle

Crop image targets vary across different stages of growth, thereby necessitating distinct tasks in CNN image processing. Figure 3 illustrates the image-processing tasks and research focuses at various stages throughout the entire crop growth cycle.

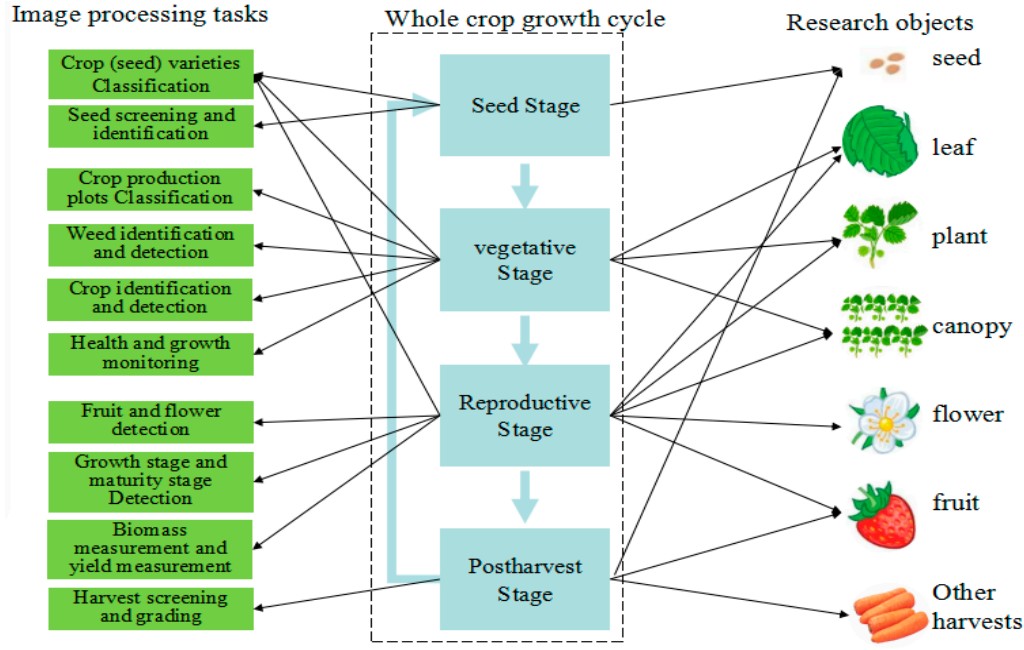

**Figure 3.** Image processing tasks and research objects of crops at different growth stages.

### 3.1. Seed Stage

Seeds, as the reproductive organs of crops, exhibit slight variations in shape, color, and texture. The collection of crop seed images was conducted within a controlled indoor environment. The image-acquisition systems predominantly utilized static setups, enabling the capture of high-quality images with minimal noise against a consistent background. Figure 4 showcases a typical image-acquisition device and sample pictures. The primary components of this setup comprise a photographing unit (RGB/multispectral camera) for image acquisition, a light source to provide optimal illumination conditions, a focusing plate to display and secure samples, a darkroom to eliminate illumination noise, and a computer equipped with software to facilitate the operation of the entire acquisition system.

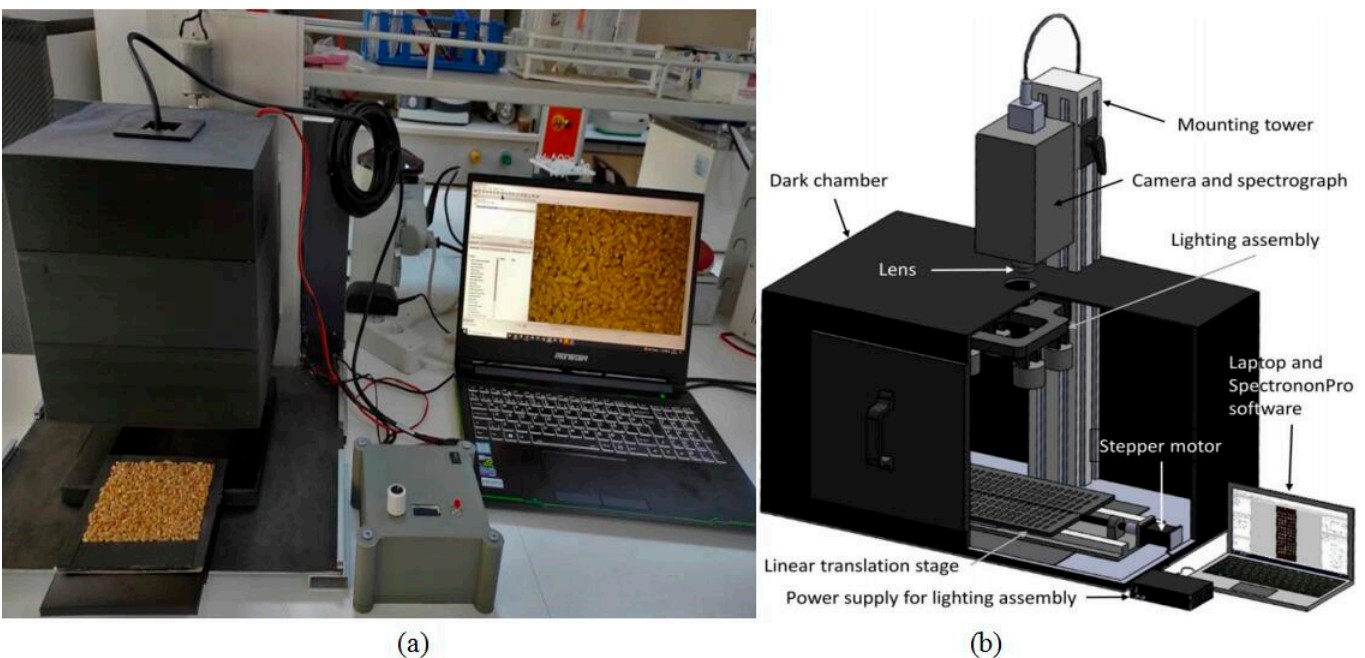

**Figure 4.** Typical image acquisition equipment. (**a**) Wheat seed. (**b**) Barley seed.

In this stage, CNN primarily finds applications in seed variety classification and seed identification. Table 1 provides a comprehensive overview of the latest research advancements in the seed stage.

### 3.1.1. Seed Variety Classification

The conventional approach to seed variety classification involves extracting phenotypic information from seed images to derive relevant features. These features are subsequently utilized as input for seed classification. For instance, Koklu et al. [1] extracted 106 phenotypic features, encompassing morphology, shape, and color attributes, from rice seed images. They then employed an artificial neural network (NN) and a deep neural network (NN) for the classification of these seeds.

Due to their small size, dense arrangement, and subtle variations, seeds are particularly suitable for classification networks. These networks have been applied or enhanced to classify different crop varieties using visible or multispectral images (MSI) of seeds. For instance, utilizing RGB images, researchers proposed the Lprtnr1, VGG, self-built CNN, and ResNet models to classify hazelnut [11], chickpea [12], rice [13], and pepper [14] seeds, achieving impressive accuracy rates of 98.63%, 94.0%, and 99.87%, respectively. The combination of MSI and CNN is a common approach. Several studies [15–19] obtained MSI of seeds and subsequently developed or improved classification networks for the classification and recognition of barley, rice, soybean, pepper, ladies finger plant, and other seed varieties. By utilizing both visible images and MSI, researchers can leverage complementary advantages, obtaining visual information and the spectral reflection characteristics of the seeds, leading to improved seed variety classification. For example, Zhou et al. [20] proposed a pixel spectral image reconstruction method based on CNN, employing visible and near-infrared (NIR) images, achieving a recognition rate of 98.15% for corn seed varieties. Furthermore, the classification network has been utilized for depth feature extraction in traditional machine-learning classification models, resulting in enhanced classification accuracy. Javanmardi et al. [21] demonstrated the superiority of CNN-based feature extraction models over traditional approaches, showcasing the accurate classification of corn seed varieties through the combination of the classification network and the artificial neural network (NN). Similarly, Unlersen et al. [22] employed seven pre-trained CNN models to extract depth features from wheat seed images, subsequently utilizing a support vector

machine for classification, achieving a classification accuracy rate of 98.1%, surpassing that of a single traditional machine-learning model.

### 3.1.2. Seed Screening and Identification

Damaged and unhealthy seeds can adversely impact crop growth and breeding processes. To address this, researchers have employed self-built or enhanced classification networks using RGB images or MSI of the same crop seeds to identify and screen damaged or imperfect seeds. For instance, five different CNN models [23], an improved VGG model [24], a self-built CNN [25], and an enhanced ResNet model [26] were utilized to identify damaged and imperfect seeds of rice, corn, and wheat, respectively. The classification network effectively screened seed phenotypes during breeding activities. Additionally, Sabadin et al. [27] capitalized on the distinction between haploid and diploid maize seed images, employing an improved CNN model. This approach successfully identified haploid seeds, providing valuable assistance in maize breeding efforts.

**Table 1.** Application progress of CNN in the seed stage.

| Application Direction | Crop Varieties | Literature | Year | Image Processing Task | Network Framework | Accuracy |
|---|---|---|---|---|---|---|
| seed variety classification | rice | [16] | 2022 | identification of rice seed varieties | GoogLeNet, ResNet | 86.08% |
| | rice | [1] | 2021 | rice variety classification | VGG 16 | 99.87% |
| | rice | [13] | 2021 | rice seed type classification | RiceNet | 100.00% |
| | maize | [20] | 2021 | maize seed identification | LeNet-5 | 98.15% |
| | maize | [21] | 2021 | corn seed classification | CNN-ANN | 98.1% |
| | wheat | [22] | 2022 | wheat varietal classification | DenseNet201 | 98.10% |
| | barley | [15] | 2021 | barley seed variety identification | CNN | 98.00% |
| | soybean | [17] | 2020 | soybean seed variety identification | | 97.20% |
| | chickpea | [12] | 2021 | chickpea situ seed variety identification | VGG16 | 94.00% |
| | pepper | [18] | 2020 | pepper seed variety discrimination | 1D-CNN | 92.60% |
| | pepper | [14] | 2021 | pepper seed classification | ResNet | 98.05% |
| | hazelnut | [11] | 2021 | variety classification in hazelnut | Lprtnr1 | 98.63% |
| | okra | [19] | 2021 | hybrid okra seed identification | CNN | 97.68% |
| seed screening and identification | rice | [23] | 2022 | milled rice grain damage classification | EfficientNet-B0 | 98.37% |
| | maize | [24] | 2022 | maize seed classification | SeedViT | 97.60% |
| | maize | [27] | 2021 | haploid maize seed identification | CNN | 97.07% |
| | wheat | [25] | 2022 | unsound wheat kernel discrimination | CNN | 96.67% |
| | grain | [26] | 2022 | bulk grain sample classification | ResNet | 98.70% |

### 3.1.3. Brief Summary

- Dataset construction: MSI and hyperspectral images (HSI) offer richer seed phenotypic characteristics compared with visible images, thus finding widespread application in CNN-based seed classification and identification. During the seed stage, the image-acquisition process becomes relatively complex. Furthermore, acquiring image samples is a time-consuming and labor-intensive task, leading most studies to rely on self-built datasets.
- Model selection: The seed stage involves relatively simple image processing tasks, with the CNN application primarily focusing on the classification network. Specifically, efforts concentrate on improving conventional and self-built classification networks. These models tend to be compact and highly efficient, with no inclusion of CNN object detection or segmentation networks within the scope of the selected literature.
- Existing problems: The utilization of standardized indoor collection environments and uniform image backgrounds result in a limited generalization ability for trained CNN models. Consequently, much of the research remains in the experimental stage and struggles to be effectively applied in real-world production scenarios. Moreover, the scarcity of high-quality seed data samples poses challenges in implementing large-scale and deep CNN models.

- Further research: The untapped potential of multispectral and hyperspectral images warrants further exploration, as they offer the ability to visualize the internal features and components of seeds in a "non-destructive" manner. By integrating these imaging techniques with breeding and genetic trait knowledge, CNN networks hold the promise of providing fast, cost-expensive, and non-destructive detection tools for breeding and seed production. This would significantly enhance the efficiency of breeding efforts.

### 3.2. Vegetative Stage

In the vegetative stage, the focus of research revolves around leaves, plants, and crop canopies as the primary image targets. Leaves serve as the central organ for crop photosynthesis, and their phenotypic characteristics reveal variations among different crop varieties. Plant and canopy images offer a wealth of phenotypic features that reflect crop varieties, growth, and nutritional status. To obtain crop images in the vegetative stage, various methods are employed. These images are typically collected from real production environments such as fields, greenhouses, and orchards. However, the backgrounds of these images tend to be complex and noisy, while lighting conditions, shooting angles, and shooting distances can also impact the quality of the images. Table 2 summarizes the latest research progress in the vegetative stage.

### 3.2.1. Crop Variety Classification

Crop variety classification and recognition using computer vision techniques is a vibrant and active research field. The PlantCLEF plant recognition challenge has been conducted consistently over several years, attracting research teams from around the world to participate and compete. The primary objective of this challenge is to identify plants on a large scale in real-world scenarios. In PlantCLEF 2017, a comprehensive collection of 200,000 images was presented, featuring 10,000 different species of herbs, trees, and other plants. These images encompassed various perspectives and plant organs, including the whole plant, fruits, leaves, flowers, stems, branches, and scanned leaves [28].

The classification networks are primarily employed for crop variety classification, where whole pictures serve as the input and a self-built or enhanced classification network is utilized to classify the crop variety. Leaves are widely utilized to differentiate between different crop varieties. For instance, Nasiri et al. [29] and Liu et al. [30] employed VGG16 and GoogLeNet, to identify grape varieties based on leaf images captured in orchards. Selvam et al. [31] achieved the accurate classification of okra varieties using okra leaf images and a CNN, leveraging features such as leaf morphological characteristics and the fingerprint function. Leaf venation structure and vein morphology are also considered significant characteristics for plant variety identification. Grinblat et al. [32] collected leaf images of three types of bean seedlings, extracted the "fingerprint" of soybean varieties using leaf vein morphology images, and successfully identified white beans, red beans, and soybeans. Vayssade et al. [33] employed a Mask-R-CNN segmentation network to segment densely planted crop leaves.

### 3.2.2. Weeds Identification and Detection

Food loss due to weeds amounts to approximately 13.2% annually [9]. Therefore, the accurate and rapid identification of weeds using machine vision enables accurate pesticide application and reduces the dosage of pesticides [7].

Classification networks play a pivotal role in distinguishing weed varieties from crops. Manikandakumar et al. [34] used two weed datasets to distinguish various weed species. By assigning distinct categories to soybean, broad-leaved clover, and soil, a deep residual CNN [35] was employed to classify weeds in soybean fields. Garibaldi-Marquez [36] achieved a higher accuracy rate in classifying corn, narrow-leaf weeds, and broadleaf weeds. By constructing a new network, VGG-Beet, Moazzam et al. [37] used an airborne multispectral camera to detect weeds in beet fields and achieved a higher

detection accuracy and a lower detection time. Similar studies have been conducted by other researchers [38,39]. The feasibility of employing deep CNNs (DCNNs) for the systematic detection of wheat broadleaf weed seedlings was systematically evaluated [40] employing models such as AlexNet, DensNet, ResNet, and VGG.

The target-detection network primarily focuses on field weed detection, aiming to identify and classify "non-crop" plants or regions within specific crop production environments. Based on the level of detection granularity, it can be categorized into weed plant and weed area detection. Weed plant detection involves training the network using "close-up" images of weeds and focuses on detecting, segmenting, and recognizing individual weed plants. Gao et al. [41] developed a compact YOLOv3 model specifically for weed detection in beet fields, achieving an accuracy rate of 82.9%. While weed detection models are typically tailored to specific crop production environments, Sapkota et al. [42] demonstrated the potential for the "migration learning" of weed identification across different crop fields, thus enhancing the universality of the weed detection model. They applied YOLOv4 and Faster R-CNN weed detection models, originally trained in cotton fields, to weed detection in soybean and corn fields. Weed area detection, on the other hand, involves capturing crop canopy images using unmanned aerial vehicles (UAVs) or other equipment, followed by the application of CNNs to identify areas with high weed density within production plots. This provides valuable information for the precise operation of agricultural machinery, such as spraying UAVs. With the use of low-altitude UAV images, an improved Faster R-CNN was utilized to detect weeds in the middle and late stages of soybean fields [43], and the generalizability of the model was found to be good. Hennessy et al. [44] proposed YOLOv3-tiny for identifying two weeds in wild blueberry fields, while YOLOv3 was effectively employed to detect weeds in alfalfa crops [45].

Segmentation networks play a crucial role in distinguishing crops and weeds from the image background, facilitating the detection of weeds within crop fields. Nasiri et al. [46] employed a U-Net architecture to achieve pixel-level semantic segmentation of beets, weeds, and soil in field images. Su et al. [47] introduced a novel data enhancement method specifically designed for semantic segmentation tasks. Their proposed method enhanced the quality and diversity of the training data, leading to improved segmentation performance.

### 3.2.3. Classification of Crop Production Plots

The classification network is extensively utilized to differentiate and map various crop plots using crop canopy images acquired from unmanned aerial vehicles (UAVs) and remote-sensing satellites. Agilandeeswari et al. [48] classified different crop plots, including corn, soybean, and lettuce, by measuring their reflectivity under visible light (Vis), NIR, and short-wave IR (SWIR). Pandey et al. [49] proposed a novel architecture called the conjugate dense network (CD-CNN), which employed RGB images captured by drones to achieve the accurate classification of diverse crop fields, yielding an accuracy rate of 96.2%. Additionally, a DCNN framework incorporating a conditional random field classifier was proposed [50], establishing a UAV hyperspectral image (WHU-Hi) dataset for precise crop plot classification. Similar studies have also been conducted by other researchers [51–56].

Segmentation networks are employed to segment and map production plots based on crop canopy images. Jayakumari et al. [57] obtained the point cloud data of cabbage, tomato, and eggplant through a high-resolution lidar and designed a DCNN model, Crop-PointNet, to semantically segment crops from a 3D perspective. Ji et al. [58] proposed a 3D FCN embedded with global pool and channel attention modules, which extracted the spatiotemporal features of different crop types from multi-temporal high-resolution satellite images. Wang et al. [59] utilized the China GF-2 remote-sensing satellite to acquire winter wheat images, and developed the RefineNet-PCCCRF model, which accurately extracted the large-scale spatial distribution of winter wheat.

### 3.2.4. Crop Identification and Detection

The accurate identification, location, and detection of crop plants or plant organs are crucial requirements for the operation of intelligent agricultural machinery and the implementation of "machine replacement" in agriculture. During the vegetative stage, the focus of crop identification and detection primarily revolves around individual plants.

Target-detection networks are employed to detect plant seedlings in the field. Tseng et al. [60] utilized migration learning techniques with EfficientDet-D0 and Faster R-CNN models to identify and detect the field rice seedlings photographed by a UAV. Aeberli et al. [61] leveraged multi-temporal UAV spectral images for the automatic detection of crop plants. Liu et al. [62] achieved the fast and automatic counting of corn seedlings using Faster R-CNN with RGB images obtained from drones. Furthermore, target-detection networks have been applied to tree mapping in forest environments. Pearse et al. [63] proposed a rapid and large-scale mapping approach for conifer seedlings using CNNs and RGB orthogonal imaging, achieving an accuracy of 98.8%.

Additionally, target-detection networks can also be combined with segmentation networks to facilitate instance segmentation based on the target location. Zhang et al. [64] first extracted the sub-image of individual target leaves from the entire plant image using Faster R-CNN and subsequently performed leaf segmentation. Wu et al. [65] collected data from apple trees in orchards using a UAV, and with the assistance of a Faster R-CNN detector and U-Net model, they successfully detected, counted, and segmented apple trees while extracting crown parameters. The Faster R-CNN detector detected apple trees, while the U-Net model performed segmentation on the detected trees.

### 3.2.5. Health and Growth Monitoring

Leaf images play a significant role in crop health diagnosis, particularly in the identification of crop diseases. The PlantVillagedataset is widely utilized for image diagnosis, providing images of healthy leaves as well as leaves affected by various diseases across 35 different crop types, all against a black background. Building upon this dataset, the AI Challenge 2018 image dataset was developed, featuring real-scene images of healthy and diseased leaves from 28 crop types, which has also gained considerable usage.

Classification networks are employed for crop state diagnosis based on leaf images. Crop growth can be monitored by classifying images of different crop growth stages. Segmented mango leaves were used for the stress recognition of various mango leaves based on a self-built CNN [66]. Similarly, A CNN model was utilized to classify medicinal crop varieties and their maturity based on leaves [67]. Gang et al. [68] employed ResNet 50 to establish an estimation model for the growth indexes (fresh and dry weights, height, leaf area, and diameter) of lettuce in a greenhouse. Another study utilized a GL-CNN model to classify red phoenix vegetables based on leaf images during the growth period [69]. Tan et al. [70] achieved the automatic detection of rice seedlings at different varieties, seedling densities, and sowing dates using EfficientnetB4, with an accuracy of 99.47%.

Target-detection networks can also contribute to crop health monitoring by assigning different labels to distinct health conditions. Yarak et al. [71] combined high-resolution images with faster R-CNN to automatically detect and classify oil palm trees and proposed a new method for their management with an accuracy of 86.96%.

Segmentation networks enable the monitoring of crop growth stages by segmenting crop canopies. For example, Tian et al. [72] monitored the growth status of corn and rice using an updated CNN structure. Zhang et al. [73] employed a segmentation network to separate lettuce plants from the background and estimated the canopy area of lettuce according to the pixel area. By establishing the relationship between images and related traits such as fresh and dry leaf weights and leaf areas, the growth stage of lettuce was effectively monitored.

**Table 2.** Application progress of CNN in the vegetative stage.

| Application Direction | Crop Varieties | Literature | Year | Image Processing Task | Network Framework | Accuracy |
|---|---|---|---|---|---|---|
| crop variety classification | grape | [30] * | 2021 | grapevine cultivar identification | GoogLeNet | 99.91% |
| | grape | [29] | 2021 | grapevine cultivar identification | VGG16 | 99.00% |
| | ladies finger plant | [31] | 2020 | ladies finger plant leaf classification | CNN | 96.00% |
| weeds identification and detection | weed | [34] | 2023 | weed classification | CNN | 98.58% |
| | weed | [42] | 2022 | weed detection | YOLOv4, Faster R-CNN | 88% |
| | weed | [47] | 2021 | crop–weed classification | | 98.51% |
| | weed | [46] | 2022 | weed recognition | U-NET | 96.06% |
| | weed | [45] | 2022 | weeds growing detection | | 98%% |
| | weed | [39] | 2020 | weed and crop recognition | GCN-ResNet-101 | 99.37% |
| | weed | [36] | 2022 | weed classification | | 97.00% |
| | weed | [40] | 2021 | broadleaf weed seedlings detection | | |
| | weed | [37] | 2021 | weeds detection | VGG16 | |
| | weed | [43] | 2020 | weed detection | Faster RCNN | 85% |
| | weed | [44] | 2021 | weed identification | YOLOv3-Tiny | 97.00% |
| | weed | [38] | 2020 | mikania micrantha kunth identifying | MmNet | 94.50% |
| | weed | [35] | 2022 | weed detection in soybean crops | DRCNN | 97.30% |
| crop production plots classification | vegetable | [52] | 2020 | 8 vegetables and 4 crops | ARCNN | 92.80% |
| | blueberries | [54] | 2020 | legacy blueberries recognition | CNN composed of eight layers | 86.00% |
| | crop | [50] | 2020 | crop identification | CNNCRF | |
| | crop | [51] | 2020 | crop classification | 2D-CNN | 86.56% |
| | wheat | [59] | 2020 | winter wheat spatial distribution | RefineNet-PCCCRF | 94.51% |
| | crop | [49] | 2022 | crop identification and classification | CD-CNN | 96.20% |
| | crop | [48] | 2022 | crop classification | | 99.35% |
| | crop | [55] | 2020 | crop classification | CNN-Transformer | |
| | crop | [53] | 2020 | crop classification | Conv1D-RF | 94.27% |
| | rice | [56] | 2020 | rice-cropping classifying | AlexNet | 94.87% |
| | vegetable | [57] | 2021 | vegetable crops object-level classification | CropPointNet | 81.00% |
| | crop | [58] | 2020 | precise crop classification | 3D FCN | 86.50% |
| crop identification and detection | tree | [63] | 2020 | tree seedlings detecting | Faster R-CNN | 97.00% |
| | rice | [60] | 2022 | rice seedling detection | EfficientDet, Faster R-CNN | 88.80% |
| | banana | [61] | 2021 | banana plants detection | | 93.00% |
| | apple | [65] | 2020 | apple tree crown extracting | Faster R-CNN | 97.10% |
| | maize | [62] | 2022 | maize seedling number estimating | Faster R-CNN | |
| | potato | [64] | 2021 | leaf detection | Faster R-CNN | 89.06% |
| | flower | [74] | 2021 | detection and location of potted flowers | YOLO V4-Tiny | 89.72% |
| health and growth monitoring | medicinal materials | [67] | 2021 | medicinal leaf species and maturity identification | CNN | 99.00% |
| | lettuce | [68] | 2022 | lettuce growth index estimation | ResNet50 | |
| | lettuce | [73] | 2020 | growth monitoring of greenhouse lettuce | CNN | 91.56% |
| | rice | [70] | 2022 | rice seedling growth stage detection | EfficientnetB4 | 99.47% |
| | gynura bicolor | [69] | 2020 | gynura bicolor growth classification | GL-CNN | 95.63% |
| | mango | [66] | 2022 | mango leaf stress identification | CNN | 98.12% |
| | oil Palm | [71] | 2021 | oil palm tree detection | Resnet-50 | 97.67% |
| | maize, rice | [72] | 2022 | corn and rice growth state recognition | | |

* The research has been applied in real scenarios.

### 3.2.6. Brief Summary

- Dataset construction: In this stage, the collection of image data is diverse, with various production scenarios, image targets, and collection devices. The majority of data are collected from real-life scenes, and there is an abundance of publicly available data resources.
- Model selection: The classification network is primarily utilized for crop variety classification, weed identification, production plot classification, and health monitoring. The target-detection network is mainly employed for plant, organ, and weed detection, crop counting, and growth stage identification. The commonly used algorithms include the YOLO series and Faster R-CNN. The segmentation network is primarily

used for separating plants or organs from the background and is applied in growth modeling. Semantic segmentation algorithms such as SegNet, Fully Convolutional Network (FCN), U-Net, DeepLab, and Global Convolutional Network are commonly used [37].

- Existing problems: Each dataset may vary in terms of collection perspective, hardware platform, collection cycle, image type, and other aspects. This leads to the limited adaptability of datasets across different tasks. The CNN-based target-detection network can effectively detect weed areas and patches, but the model size and operational efficiency can be limiting factors. Additionally, crop production land classification based on remote sensing images may suffer from low image resolution and small feature size, which can impact the accuracy of CNN classification and segmentation networks.

- Further research: To enhance the generalization ability of models, multiple publicly available image datasets can be used for training and improvement. Transfer learning techniques can also be applied to leverage knowledge from one image processing task to another. In the context of weed identification, the target-detection and segmentation networks hold significant value. Instead of focusing on identifying specific weed species, farmers are more interested in identifying whether a plant is a weed and its location. In real-field operations, an "exclusion strategy" can be considered, where green plants not identified as "target crops" are assumed to be "weeds". Leveraging the regular spatial attributes of mechanized planting, weeds (areas) can be accurately identified and located with reduced computational requirements.

*3.3. Reproductive Stage*

The reproductive stage encompasses various image objects, such as leaves, plants, canopies, flowers, and fruits. Flowers and fruits, being the reproductive organs of crops, exhibit distinctive morphological and color characteristics, making them the primary focus of target detection. Similar to the vegetative stage, the image-acquisition process in this stage involves a complex environment, diverse methods, and variations in image quality. Most of the pictures are in RGB/RGB-D format; HSI and MSI applications are rare.

This stage attracts mechanized picking, which requires fast and accurate image recognition. In addition to conventional equipment, such as cameras, UAVs, and mobile phones, various special image-acquisition devices have been developed to assist in mechanized harvesting (Figure 5). The acquisition devices generally include a photography unit (camera) to collect images, a moving unit that includes driving motor and moving device, which moves autonomously and collect images and videos continuously and efficiently, as well as a computer with software to control the operation of the entire acquisition system. Table 3 summarizes the latest research progress in the reproductive stage.

3.3.1. Crop Variety Classification

In this stage, classification networks play a crucial role in categorizing crop varieties based on fruits, flowers, and other organs [75–78]. Notably, improved versions of AlexNet, GoogLeNet, and ResNet have been employed in grape variety recognition using grape cluster fruit images [75]. Wang et al. [76] proposed a dynamic ensemble selection method based on MobileNet, achieving an impressive accuracy of 95.5% in classifying five different types of flowers.

3.3.2. Fruit and Flower Detection

Classification networks are employed for fruit detection in various applications. Aquino et al. [79] successfully recognized fruits on olive trees by analyzing cut sub-images to determine the presence of fruit targets.

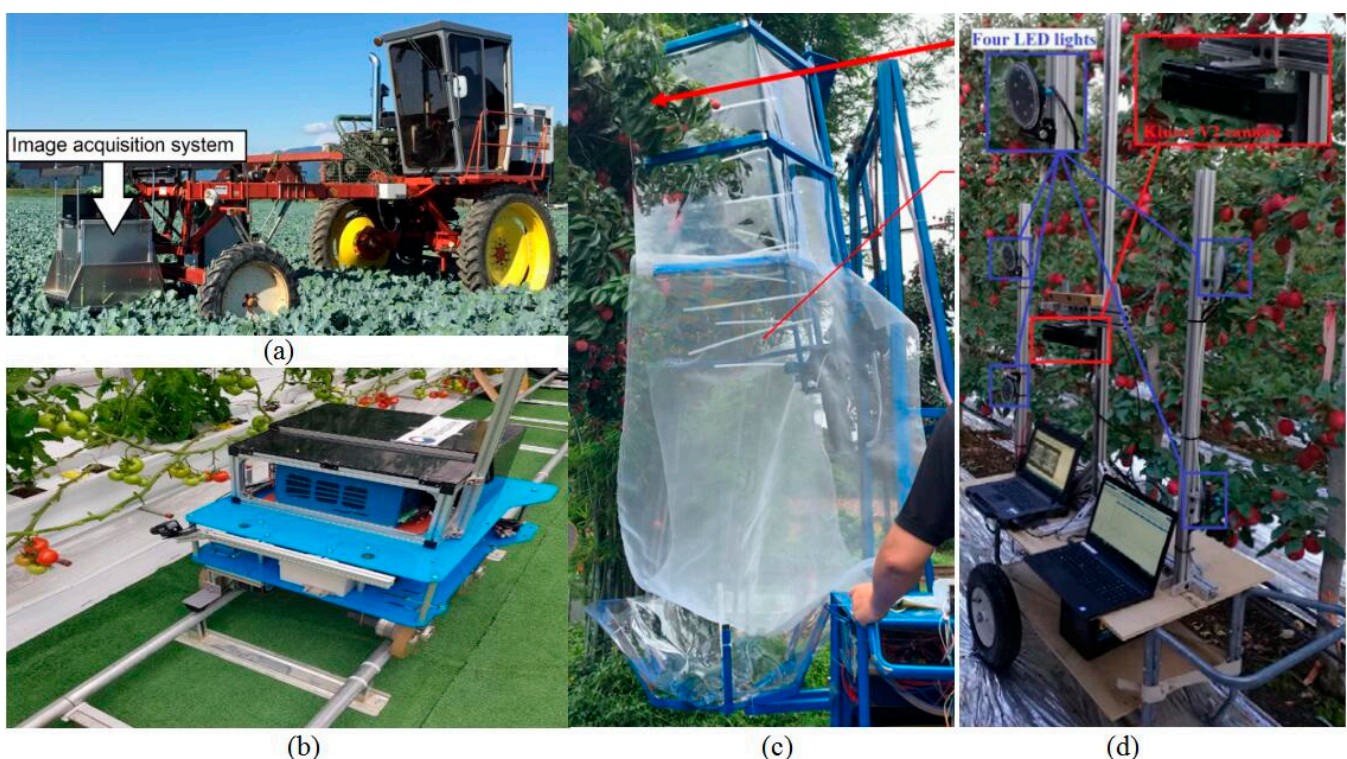

**Figure 5.** Some special image acquisition devices in the reproductive stage. (**a**) Broccoli detection. (**b**) Tomato recognition. (**c**) Litchi harvest. (**d**) Apple detection.

Target-detection networks play a vital role in fruit and crop detection, particularly in the context of increasing demand for mechanized harvesting. The accurate identification and detection of crops fruits or harvestable parts can improve the performance of picking robots. Fu et al. [80] proposed a fast and accurate target-detection algorithm, YOLOv3-Tiny, which automatically detected kiwifruit in the orchards and improved the applicability of real-time detection. Similarly, a YOLOv3-Tiny litchi network model was proposed [81], accurately identifying the distribution of fruits on litchi trees and providing a spatial map for precise mechanical picking. Chen et al. [82] improved the YOLOv3 cherry tomato detection algorithm, achieving efficient detection and recognition of small target tomato fruits with an average processing time of 58 ms. Additionally, Faster R-CNN [83,84], MobileNetV2 [85], ResNet [86], and SwinGD [87] have been utilized for apples, kiwifruit, tomato, and grape detection, respectively. The accurate detection of crop flowers is helpful in accurate pollination to improve the pollination rate. The accurate determination of spikelet flowering time is vital for the timely pollination of hybrid rice seed production [88]. Liu et al. [89] realized corn tassel detection based on UAV images and Fast R-CNN. Chandra et al. [90] proposed a reliable spike detector for wheat and sorghum using Faster R-CNN. Mask R-CNN was used to segment, identify, and count grape inflorescence samples [91]. The real-time detection of apple flowers [92], outdoor strawberry flowers [93], and apple flower buds [94] was achieved using YOLOv4 and Faster R-CNN, respectively.

Segmentation networks also play a crucial role in fruit detection. Gene-Mola et al. [95] proposed an apple detection and 3D positioning method based on Mask R-CNN instance segmentation networks and moving structure photogrammetry. By projecting a 2D segmentation mask onto a 3D point cloud, the detection accuracy of apple fruits on trees was improved by 9.5%. Xu et al. [96] utilized an improved Mask R-CNN model for instance segmentation to accurately segment cherry tomatoes. The combination of the Otsu binarization algorithm and CNN efficiently enabled the recognition of coffee flowers [97].

### 3.3.3. Growth Stage and Maturity

Classification networks can also be utilized to detect crop growth stages and maturity by assigning different category labels to fruits or flowers at various stages of growth and maturity. A DCNN was employed to extract features, generate a feature transformation matrix, reason target markers [98], and recognize cotton boll maturity states. Khosravi et al. [99] successfully recognized the recognition of four mature stages of two olive varieties based on RGB images and DCNN. Zheng et al. [100] proposed a strawberry appearance quality recognition method called Swin-MLP, which combined the Swin Transformer and multilayer perceptron to achieve accurate recognition.

Target-detection networks are employed to assign different "category labels" to different growth stages of the same crop and to identify the growth stages of target individuals in the image. Psiroukis et al. [101] effectively classified and detected the three maturity levels of broccoli crops by constructing Faster R-CNN and CenterNet. In a hydroponic greenhouse, a fast R-CNN was improved to monitor the maturity of tomato fruit [102]; Hsieh et al. [103] proposed a beef tomato fruit maturity and position recognition model based on R-CNN and binocular imaging technology. Sweet pepper development stages were detected by integrating CNN and MLP models [104].

Segmentation networks are also utilized to assess the state of flowers or fruits. An improved Mask R-CNN algorithm, MASU R-CNN, was proposed to recognize and segment apple blossoms in three states, achieving a high detection rate of 96.43% [105]. EfficientNetB0 was employed to detect the maturity of fresh oil palm fruit [106].

### 3.3.4. Biomass and Yield Measurement

Classification networks can be employed to estimate crop biomass and yield. Phenotypic characteristics such as above-ground biomass and leaf area during crop growth are key indicators of crop growth status. Oliveira et al. [107] improved a classification network based on UAV-RGB images, enabling the effective prediction of the dry matter yield of pastures.

Target-detection networks play a crucial role in identifying and detecting harvestable crops in images, thus enabling yield prediction through counting. Lu et al. [108] enhanced the field prediction of the YOLOv3 of soybean yield based on pod and leaf images; the soyabean leaves and pods were identified and counted, respectively. Similarly, Faster R-CNN with Inception v2 was used to recognize and count three types of fruits [109]. The accurate monitoring of crop growth and development stages is of great significance to fine crop management and the precise operation of agricultural machinery. Target-detection networks are also utilized for the non-destructive detection of crop growth stages. Kartal et al. [110] counted mung bean and chickpea plants using 3D laser scanning. Wang et al. [111] detected anomalies in tomato plants through improvements to the YOLO-Dense algorithm. Xu et al. [112] detected and counted corn plants by acquiring corn leaf images using a UAV.

Segmentation networks are mainly utilized for biomass estimation based on plant partition and yield estimation based on fruit partition. Safonova et al. [113] utilized Mask R-CNN to segment olive crowns based on UAV-RGB images, and the accuracy in estimating the individual volume of the crown reached 82%. Lin et al. [114] integrated image segmentation with CNN models to develop a pipeline for sorghum ear detection and counting.

The combination of target-detection and segmentation networks finds wide application in target recognition for harvesting robots. Blok et al. [115] designed an image-acquisition robot for broccoli. By acquiring RGB and depth images, the count of broccoli was realized by Mask R-CNN. A fuzzy Mask R-CNN model was proposed to identify the maturity of cherry tomatoes, with which detection and segmentation networks were used [116]. Furthermore, a vision detector model for harvesting robots based on Mask R-CNN was proposed, achieving a high detection rate of 97.31% for overlapping apples [117].

**Table 3.** Recent progress of CNN application in the reproductive stage.

| Application Direction | Crop Varieties | Literature | Year | Image Processing Task | Network Framework | Accuracy |
|---|---|---|---|---|---|---|
| crop variety classification | grape | [75] | 2021 | grape variety identification | AlexNetGoogLeNet | 96.90% |
| | tree | [76] | 2021 | pollen monitoring | SimpleModel | 97.88% |
| | flower | [77] | 2022 | flower classification | MobileNet | 95.50% |
| | tree | [78] | 2021 | tree species classification | ResNet-18 | |
| fruit and flower detection | kiwifruit | [80] | 2021 | kiwifruit detection | DY3TNet | 90.05% |
| | litchi | [81] * | 2022 | litchi harvester | YOLOv3-tiny litchi | 87.43% |
| | cherry | [82] | 2021 | cherry tomatoes detection | Yolov3-DPN | 94.29% |
| | apple | [83] | 2020 | apple detection | Faster R-CNN with ZFNet and VGG16 | 89.30% |
| | apple | [84] | 2020 | apple detection | Faster R-CNN with VGG16 | 87.90% |
| | kiwifruit | [85] * | 2020 | kiwifruit detection | MobileNetV2, InceptionV3 | 90.80% |
| | tomato | [86] | 2020 | Immature tomatoes detection | Resnet-101 | 87.83% |
| | olive | [79] | 2020 | olive fruit identification | | 98.22% |
| | fruit | [95] | 2020 | fruit detection | | 88.10% |
| | grape | [87] | 2021 | grape bunch detection | SwinGD | 91.50% |
| | tomato | [96] | 2022 | cherry tomato recognition | Mask R-CNN | 93.76% |
| | chickpeas | [110] | 2021 | plant detection and automate counting | | 93.18% |
| | maize | [89] | 2020 | maize tassel detection | Faster R-CNN | 94.99% |
| | apple | [92] | 2020 | apple flower detection | YOLO v4 | 97.31% |
| | coffee | [97] | 2020 | coffee flower identification | VGGNet | 80.00% |
| | strawberry | [93] | 2020 | strawberry flower detection | Faster R-CNN | 86.10% |
| | apple | [94] | 2022 | apple flower bud classification | YOLOv4 | |
| growth stage and maturity detection | olive | [113] | 2021 | olive tree biovolume | Mask R-CNN | 82% |
| | guineagrass | [107] | 2021 | estimate dry matter yield of guineagrass | AlexNet, ResNeXt50 | |
| | soybean | [108] | 2022 | soybean yield prediction | YOLO v3 | 90.30% |
| | tomato | [111] | 2021 | tomato anomalies | YOLO-Dense | 96.41% |
| | tomato | [116] | 2020 | tomato ripeness identification | Fuzzy Mask R-CNN | 98.00% |
| | broccoli | [115] * | 2020 | broccoli head detection | Mask R-CNN | 98.70% |
| | sorghum panicle | [114] | 2020 | sorghum panicle detection and counting | U-NET CNN | 95.50% |
| | apple | [117] | 2020 | overlapped fruits detection and segmentation | mask R-CNN | 97.31% |
| biomass and yield measurement | tomato | [102] | 2021 | tomato fruit monitoring | Faster R-CNN | 90.20% |
| | olive | [99] | 2021 | olive ripening recognition | M2 + NewBN | 91.91% |
| | sweet pepper | [104] | 2021 | sweet pepper development stage prediction | YOLO v5 | 77.00% |
| | cotton | [98] | 2020 | cotton boll status identification | NCADA | 86.40% |
| | broccoli | [101] | 2022 | broccoli maturity classification | Faster R-CNN, CenterNet | 80.00% |
| | tomato | [103] | 2021 | tomato fruit location identification | ResNet-101, Mask R-CNN | 95.00% |
| | strawberry | [100] | 2022 | strawberry appearance quality identification | Swin-MLP | 98.45% |
| | oil palm | [106] * | 2021 | oil palm ripeness classification | EfficientNetB0 | 89.30% |
| | apple | [105] | 2020 | apple flowers instance segmentation | MASU R-CNN | 96.43% |

* The research has been applied in real scenarios.

### 3.3.5. Brief Summary

- Dataset construction: Similar to the vegetative stage, there is an abundance of image resources available for fruit detection. Specifically, there are ample resources for detecting fruits in images.
- Model selection: The focus of CNN applications in this stage is on CNN object detection networks, with the Faster R-CNN and YOLO algorithms being popular choices. These algorithms primarily enable the detection of fruits and flowers, and also facilitate crop yield or maturity detection. Classification networks are mainly utilized for classifying flowers and fruits. Segmentation networks are employed, for instance in the segmentation of crop fruits, flowers, and plants, with the Mask R-CNN framework being a popular choice.

- Existing problems: While there are numerous image acquisition devices available, the combination of multiple acquisition devices is not commonly practiced. The reported accuracy of the existing research is mostly above 80%, but this is limited to specific datasets. When trained models are deployed in real production scenarios, their accuracy and speed often fall below the benchmark.
- Further research: In crop variety classification, exploring the use of organ images other than leaves, flowers, and fruits for classification can be attempted. For large-scale planting, using whole plant images for classification is recommended during the seedling stage or when plants are independent. When plants are overlapping or densely planted, it is recommended to classify crops using specific and distinct organs such as flowers, ears, and fruits. Leveraging CNN object detection networks to identify fruits at different maturity levels and integrating them into hardware devices can enable precise mechanical picking in batches and stages. Instance segmentation based on CNN classification networks can achieve the precise segmentation of fruit contours, providing precise targeting for mechanical operations.

### 3.4. Postharvest Stage

In this stage, the image target is the crop harvest, which is separated from the plant. Consequently, the images are typically collected in artificial environments such as laboratories and warehouses with uniform backgrounds and low noise. Similar to the seed stage, some static image-acquisition devices are commonly employed. However, there is a difference in the postharvest stage, as it involves commercial agricultural products, and the screening and grading of some harvests should be carried out in batches under dynamic scenes to improve the efficiency of postharvest grading. A typical image-acquisition device in the postharvest stage is shown in Figure 6.

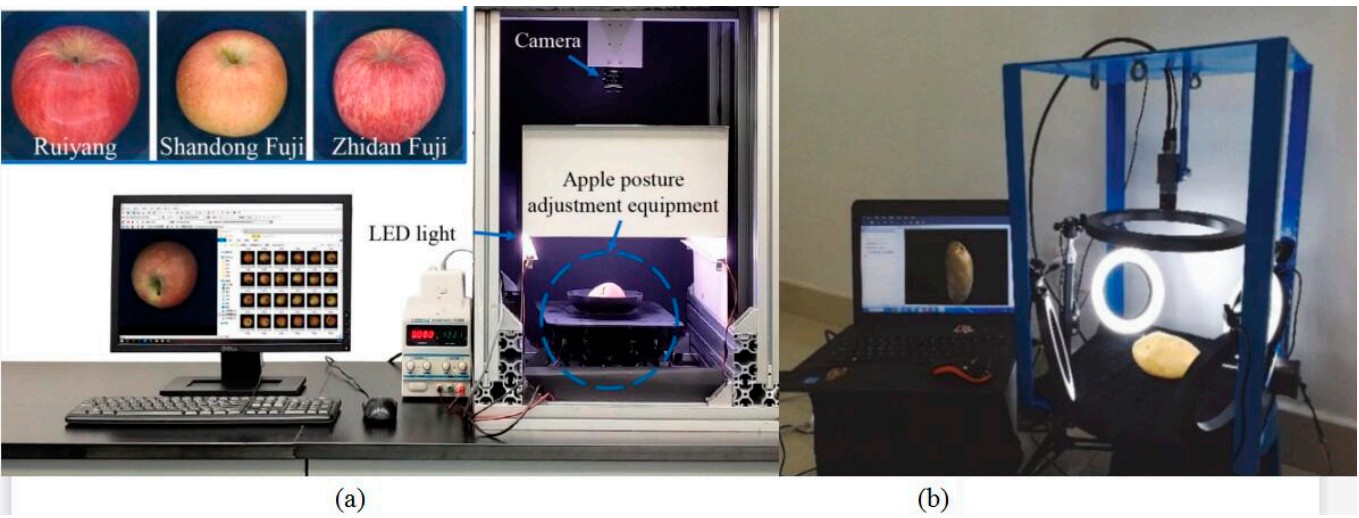

**Figure 6.** Typical image acquisition equipment in the postharvest stage. (**a**) Apple image acquisition system. (**b**) Potato image acquisition system.

Table 4 lists the latest research progress in the postharvest stage.

### 3.4.1. Harvests Screening and Grading

Grading crop harvests based on their appearance is crucial for achieving high quality and prices, which enhances the viability of a farm. Classification networks are utilized to screen and classify crop harvests [118], providing a non-destructive and fast classification method for agricultural products. Mahmood et al. [119] enhanced AlexNet and VGG16 to classify the maturity of jujube fruit by transfer learning. Momeny et al. [120] improved CNN to detect and grade the appearance of cherry fruit and enhanced the accuracy of

the algorithm. Iqbal et al. [121] automatically classified eight mango varieties based on VGG16, ResNet152, and Inception v3. A new multi-view spatial network was developed to address apple grading challenges [122]. A lightweight model, CarrotNet, based on machine vision and DCNN was proposed to classify carrots [123]. Similar studies have also been conducted [118,124–128].

Detection networks are employed to detect the special characteristics that impact the quality and grading of harvests, such as sprouts, scars, and gaps. Wang et al. [129] developed an automatic loading system for apple stem/calyx real-time recognition based on YOLO-v5, and the recognition accuracy reached 94%. A Faster R-CNN based on ResNet50 was constructed to detect potato buds [130]. Khaki et al. [131] designed a detection and counting algorithm using a sliding window to count grains on corn cobs.

**Table 4.** Application progress of CNN in the postharvest stage.

| Application Direction | Crop Varieties | Literature | Year | Image Processing Task | Network Framework | Accuracy |
|---|---|---|---|---|---|---|
| harvests screening and grading | apple | [122] | 2022 | apple quality grading | Multi-View Spatial Network | 99.23% |
| | carrot | [123] | 2021 | detecting defective carrots | CarrotNet | 97.04% |
| | potato | [127] | 2021 | potato detecting | | 97% |
| | lemon | [124] | 2020 | sour lemon classification | 16–19 layer CNN | 100% |
| | nut | [118] | 2021 | nuts quality estimation | CNN | 93.48% |
| | cherry | [120] | 2020 | cherry classification | CNN | 99.40% |
| | ginseng | [128] | 2021 | ginseng sprout quality prediction | ResNet152 V2 | 80% |
| | persimmon | [126] | 2022 | persimmon fruit prediction | VGG16, ResNet50, InceptionsV3 | 85% |
| | apple | [125] | 2021 | apple quality identification | CNN | 95.33% |
| | potato | [130] | 2020 | potato bud detection | Faster R-CNN | 97.71% |
| | mangoe | [121] | 2022 | mangoes classification and grading | InceptionV3 | 99.2% |
| | apple | [129] | 2022 | apple stem/calyx recognition | YOLO-v5 | 94% |
| | jujube fruit | [119] | 2022 | jujube fruit classification | AlexNet, VGG16 | 99.17% |
| | corn | [131] | 2020 | corn kernel detection and counting | CNN | |

### 3.4.2. Brief Summary

- Dataset construction: The dataset construction in this stage is similar to the seed stage, where efforts are made to collect and curate relevant image data.
- Model selection: The model selection in this stage is also similar to the seed stage. The main image-processing tasks involve the screening and grading of harvested crops, and most CNN applications focus on the utilization of classification and target-detection networks. Additionally, the application of hyperspectral imaging (HSI) and multispectral imaging (MSI) techniques is also observed in some studies [22,115].
- Existing problems: Similar to the seed stage, there are existing challenges and limitations in this stage that need to be addressed.
- Further research: Future research endeavors can explore the application of multispectral and hyperspectral images for batch agricultural product detection. By leveraging these advanced imaging techniques, it becomes possible to achieve the early identification of internal damage, diseases, and insect pests in agricultural products. This can significantly contribute to improving the quality assurance and screening processes of agricultural products.

## 4. Discussion

### 4.1. Self-Built Network

Self-built CNN classification networks with fewer layers have many applications, particularly in crop classification based on spectral images, including MSI, HSI, and NIR. Taner et al. [11] designed a novel CNN model, Lprtnr1, which consists of one input layer, four convolution layers, one pooling layer, one full connection layer, and one output layer. Khosravi et al. [99] constructed a new CNN for identifying the mature stage of olives on branches, which consists of three groups of convolution blocks and GAP layers and

two fully connected layers. Moazzam et al. [37] developed a new VGG-Beet network for weed identification, which is a modified VGG16 model with 11 convolution layers. Several studies are available on self-built CNN classification models [13,15,18,38,121,123,132]. The possible reasons are as follows: the classification network with fewer layers has fewer parameters and low computation, improving the operation efficiency of the model. This is particularly beneficial for MSI and HSI images with rich spectral information, as even simple models are challenging to over-fit.

### 4.2. Special Image Acquisition Device

Various image-acquisition devices have been developed for crop image classification and recognition tasks. Alongside cameras and mobile phones, UAVs and remote-sensing satellites are the most used image-acquisition devices. Multi-temporal UAVs offer advantages such as low cost, high flexibility, abundant phenological information, and real-time data acquisition, making them a crucial data source for agricultural monitoring [42,43,52,77,101,107,113,133]. They are suitable for various open-air agricultural production scenes, including fields, orchards, and forests. Currently, remote-sensing satellites provide images with wide coverage, stable time sequences, and high spatial and temporal resolutions, making them ideal for agricultural land mapping [56,58,59,134]. The more commonly used satellites are the Sentinel-2 satellite of Europe and the GaoFeng (GF) series satellite of China. Some special image-acquisition devices are also of significance. For example, Sabanci et al. [14] used Epson flat scanner to capture images of pepper seeds on a black background. Hennessy et al. [44] captured pollen images through a microscope in the laboratory. Hsieh et al. [103] employed binocular imaging technology to collect tomato fruit images from multiple angles and realized position recognition. The terrestrial laser scanner can automatically, accurately, and efficiently obtain the 3D geographical coordinates and stereo-image data of the research object. Jayakumari et al. [57] utilized a terrestrial laser scanner installed on a movable and height-adjustable tripod to obtain 3D point cloud data and constructed the PointNet model to classify cabbage, tomato, and eggplant. Tian et al. [72] integrated a laser rangefinder into a three-axis PTZ and dynamically scanned the observation area to acquire the position point set of rice and corn. Serkan et al. [110] scanned 3D images of mung bean and chickpea crops using a PlantEye laser scanner.

### 4.3. Special Objects

Most of the research in this field focuses on leaves, flowers, fruits, seeds, plants, and canopies as the primary objects of study. However, there are also studies that involve unique crop organs. For instance, bark [135] and pollen [78] were utilized to classify tree species. Tillering number is an important agronomic trait that affects rice yields. Rice harvest cross-section images have been employed to automatically detect rice tillering, which is an important agronomic trait that affects rice yield [136]. The detection of some unique objects can provide targets for mechanical operations, such as grape-picking robots, apple automatic loading systems, and farming agricultural machinery. Targets such as grape stems [137], apple stems [129], and rice seedling rows [138] have been successfully detected. While most studies focus on a single target object, using multiple target objects can lead to more generalized crop classification and recognition. For example, a soybean yield prediction method was proposed by simultaneously counting soybean pods and leaves [108]. Wheat plant and seed images were both used to realize variety classification [139]. Multi-target detection enables more accurate operation targets for picking robots. Tomato fruit strings and stems were simultaneously segmented to provide targeting for machine picking [96]. The multi-class target detection of apple trunk and fruit was used to predict the vibration position of the self-shaking apple harvesting robot [140]. Similarly, there have been studies that simultaneously detect the fruit and branches of citrus plants [141].

*4.4. Multimodal Data*

The most prevalent image pattern currently used is the 2D RGB image. Compared with RGB images, MSI provides richer images and spectral information. Table 5 lists some publicly available image datasets. In addition to a single image type, there is a growing trend towards the fusion of multi-modal data. For instance, [83,96,140] detected and located tomatoes and apples, respectively, by fusing RGB and depth images. Gene-Mola et al. [95] realized apple detection and 3D positioning by projecting a 2D partition mask onto a 3D point cloud. [68,73] segmented canopies and estimated the growth index of lettuce, respectively, through RGB and depth image fusion. Agilandeeswari et al. [48] classified crop plots by fusing and measuring Vis, NIR, and SWIR wavelengths.

**Table 5.** Partially available public image datasets.

| Datasets Name | Data Volume | Obtain Address |
|---|---|---|
| global wheat detection | 4700 | https://www.kaggle.com/c/global-wheat-detection/data (accessed on 12 January 2023) |
| flower recognition dataset | 4242 | https://www.kaggle.com/datasets/alxmamaev/flowers-recognition (accessed on 19 December 2022) |
| pest and disease library | 17,624 | http://www.icgroupcas.cn/website_bchtk/index.html (accessed on 5 January 2023) |
| AI Challenger 2018 | 50,000 | https://aistudio.baidu.com/aistudio/datasetdetail/76075 (accessed on 16 January 2023) |
| plant village | 54,303 | https://data.mendeley.com/datasets/tywbtsjrjv/1 (accessed on 5 January 2023) |
| lettuce growth images | 388 | https://doi.org/10.4121/15023088.v1 (accessed on 25 January 2023) |
| ABCPollen dataset | 1274 | http://kzmi.up.lublin.pl/~ekubera/ABCPollen.zip (accessed on 19 December 2022) |
| WGISD (grape) | 300 | https://github.com/thsant/wgisd (accessed on 22 January 2023) |
| hyperspectral remote sensing scenes | | http://www.ehu.eus/ccwintco/index.php/Hyperspectral_Remote_Sensing_Scenes (accessed on 15 January 2023) |
| weed dataset | | http://agritech.tnau.ac.in/agriculture/agri_weedmgt_fieldcrops.html (accessed on 19 December 2022) |
| ICAR-DWR database (weed) | | http://weedid.dwr.org.in/ (accessed on 19 December 2022) |
| CWD30 dataset (weed) | 219,770 | https://arxiv.org/abs/2305.10084 (accessed on 17 May 2023) |

*4.5. Cross Stage*

Several research explorations have utilized multi-time sequence images to investigate the cross-growth stages of crops. Abdalla et al. [142] successfully combined convolutional neural networks (CNN) with long short-term memory (LSTM) in the classification of rapeseed nutritional status. Their model exhibited good generalization performance across different datasets. In a study by Trevisan et al. [133], a UAV was employed to capture soybean growth images over three growing seasons. By constructing a CNN, they achieved high-throughput phenotypic analysis of soybean maturity. Another noteworthy research contribution was the establishment of a CNN-based model for male rapeseed plant recognition, enabling the segmentation of male rapeseed plants from complex backgrounds [143]. Gao et al. [139] proposed CMPNet, a wheat classification model based on ResNet and SENet. By incorporating images from the tillering stage, flowering stage, and seed stage, they significantly improved the classification accuracy of wheat varieties. Furthermore, a corn yield estimation model based on spectral and color images was introduced [144]. Additionally, a highly efficient deep convolutional neural network (DCNN) structure was proposed for detecting the development stage of rice using handheld cameras [145]. The proposed model achieved a detection accuracy of 91.3% by utilizing multiple views.

*4.6. Application Deployment*

Despite the majority of studies still being in the experimental phase, there is an encouraging increase in research with practical applications, yielding tangible value. For instance, Blok et al. [115] set up a cauliflower image-acquisition robot, which automatically detects and identifies broccoli heads and picks them by walking in the fields. Through cost accounting, picking robots have considerable economic benefits over manual picking.

Massah et al. [146] designed and developed a crawler kiwifruit yield estimation robot utilizing machine vision technology. Li et al. [81] conducted field experiments with a column comb-type litchi harvesting robot. Furthermore, numerous mobile applications have been developed for various purposes, such as palm oil fruit maturity classification [106], grape variety classification [30], automated rice tiller detection [136], and kiwifruit detection [85], to name a few. These practical applications highlight the significant progress in the field.

## 5. Research Prospect

To enhance the practical application of these research findings, it is crucial to transition from experimental studies to real-world agricultural production scenarios. This entails developing corresponding software by integrating improved and trained models to serve agricultural operations in practical settings. One of the reasons for the gap between experiments and application lies in the complexity of agricultural production environments, which often differ from the idealized conditions under which sample images are collected. To address this, efforts should be made to improve the collection of sample images in real operation scenes.

In terms of algorithmic applications, it is important to diversify the choice of algorithms in crop classification and recognition. While studies in recent years have predominantly focused on algorithms such as VGG, AlexNet, and ResNet in classification networks, YOLO series, Fast R-CNN, and Mask R-CNN in target-detection networks, and instance segmentation algorithms based on Mask R-CNN in segmentation networks, there is potential for exploring other deep learning models such as RNN and LSTM. Leveraging the advantages of RNN and LSTM in processing time-series data, the combination of multi-time-series images and CNN can enable the classification, recognition, and prediction of crop phenotypes in the temporal dimension. LSTM, in particular, will play a significant role in time-series analysis in precision agriculture [52,122]. Additionally, considering a CNN as the pre-feature extraction module of the traditional machine-learning model can prove to be a promising approach.

The performance of CNN models is heavily influenced by the number of sample images, and thus it is essential to increase the quantity of available samples. For complex agricultural scenes, achieving acceptable target-detection results often requires at least 3000 to 4000 marked samples per class [97]. However, among more than 200 related studies conducted between 2020 and 2023, the number of image samples exceeded 20,000 in less than 10% of the cases. Data augmentation techniques can help mitigate overfitting, improve model robustness, and compensate for the limited original data by generating additional training samples. Countermeasure networks can also be employed to address the challenge of insufficient sample sizes, allowing for the manual synthesis of more samples based on existing ones [105,147]. In addition, large-scale, diverse, holistic, and hierarchical datasets would facilitate the development of more accurate, robust, and generalizable deep learning models [148].

Furthermore, the combined application of 3D crop models and multi-modal data can provide more comprehensive information on crop growth. By constructing 3D models of crops and incorporating environmental parameters (such as temperature, humidity, and light intensity), as well as physiological parameters (such as water content, photosynthetic efficiency, and chlorophyll content), a multi-modal crop model based on CNN can facilitate more in-depth and comprehensive research on crop phenotypes. This integrated approach enables a more accurate description of crop growth dynamics.

## 6. Conclusions

Convolutional Neural Networks (CNNs) have emerged as the dominant framework within deep learning models for agricultural image processing, playing a crucial role in advancing precision agriculture. This comprehensive study presents a systematic overview of CNN applications across four stages of the whole crop growth cycle: the seed stage, vegetative stage, reproductive stage, and postharvest stage. By reviewing and comparing

current research, this article offers insights into the progress made in CNN-based image processing tasks, research objectives, algorithm selection, and image acquisition equipment. Furthermore, it explores future directions for the development of CNN applications in agriculture. This study also highlights common challenges and unique applications within this field, providing valuable references for future research endeavors in this domain.

**Author Contributions:** F.Y., Q.Z. and Y.M. analyzed the data, prepared the tables, and drafted the manuscript; J.X. and M.W. designed the project and finalized the manuscript; R.L., X.L., Y.P., Y.N., Z.T. and H.Z. assisted with reference collection and the reorganization and partial data analysis. All authors have read and agreed to the published version of the manuscript.

**Funding:** This research was funded by the Beijing Innovation Consortium of Agriculture Research System (CN) (BAIC10-2023) and the Youth Fund of Beijing Academy of Agriculture and Forestry Sciences (CN) (QNJJ202213).

**Data Availability Statement:** The data presented in this study are available on request from the corresponding author.

**Acknowledgments:** The authors would like to acknowledge the anonymous reviewers for their valuable comments and members of the editorial team for their careful proofing.

**Conflicts of Interest:** The authors declared that there is no conflict of interest.

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
