# Peer review of "Progress in the Application of CNN-Based Image Classification and Recognition in Whole Crop Growth Cycles"

_remotesensing, doi:10.3390/rs15122988_

Round 1

Reviewer 1 Report

The authors had a good idea to summarize the progress on the CNN application in many fields during crop growing season. The current version may give a general picture of classification and identification with CNN algorithm in various aspect of crop production. But the key points on the real progress that CNN were used were still missing. So I would suggest the authors give the academic comments on each part and help the readers understand what questions were well addressed and what else still needs further investigation as well as which algorithm is the best choice in comparison. The topics of this article suits for more Sensor than Remote Sensing.  

I would suggest the authors consult the English-speaking people to polish the English. The current style is good to read and understand but not native style.  

Author Response

Thank you very much for your valuable feedback on the manuscript, which will make it more comprehensive and rigorous. We have made revisions and improvements to the manuscript based on each of your proposed revisions. The details are as follows:

Point 1: I would suggest the authors give the academic comments on each part and help the readers understand what questions were well addressed and what else still needs further investigation as well as which algorithm is the best choice in comparison.

Response 1: As per your suggestion, we added "Brief Summaries" to different crop growth stages. It summarizes and evaluates the existing reference schemes, existing problems, datasets construction, and algorithm selection for CNN applications at each stage of crop growth, which will help readers better understand the research progress.

Point 2: The topics of this article suits for more Sensor than Remote Sensing.

Response 2: Thank you very much for your suggestion. We believe that the topics of the manuscript are also suitable for Remote Sensing, especially in its "Deep and Machine Learning Applications in Remote Sensing Data" special section.

Point 3: I would suggest the authors consult the English-speaking people to polish the English. The current style is good to read and understand but not native style.

Response 3: Based on your suggestion, we have polished the English version of the manuscript.

Reviewer 2 Report

Reviewers Comments

The review study presented in manuscript id: remotesensing-2392066, titled “Progress in the Application of CNN-Based Image Classification and Recognition in Whole Crop Growth Cycles,” is deemed interesting and relevant in the current scenario. However, there are a few flaws that need to be addressed, hence a minor revision is being suggested. Here are some specific comments to help improve the manuscript:

1.      The authors must check for unnecessary capitalization throughout the manuscript and revise accordingly.

2.      The abstract needs to be enriched by adding 1-2 sentences highlighting the need for the review and 1-2 sentences presenting the adopted review methodology.

3.      The formulation of the research gap and review questions in the introduction (line no. 52-61) needs improvement. The authors should refer to more literature published in the area to properly formulate the research gap and review questions. The following articles can be used as references:

(i)                 https://doi.org/10.3390/rs14030559

(ii)              https://doi.org/10.1016/j.sciaf.2023.e01577

(iii)            https://doi.org/10.3390/s22197384

4.      The conclusion is too long and needs to be rewritten to highlight the major findings of the review study in a concise manner (maximum of 200 words).

I wish authors a great success.

Author Response

Thank you very much for your valuable feedback on the manuscript, which will make it more comprehensive and rigorous. We have made revisions and improvements to the manuscript based on each of your proposed revisions. The details are as follows:

Point 1: "The authors must check for unnecessary capitalization throughout the manuscript and revise accordingly."

Response 1: The manuscript has been read through and all sentences have been modified to minimize unnecessary capital letters. See the text for details.

Point 2: The abstract needs to be enriched by adding 1-2 sentences highlighting the need for the review and 1-2 sentences presenting the adopted review methodology.

Response 2: Based on your suggestion, we have added the necessity and methods of review in the abstract, as detailed in the abstract section.

Point 3: The formulation of the research gap and review questions in the introduction (line no. 52-61) needs improvement. The authors should refer to more literature published in the area to properly formulate the research gap and review questions. The following articles can be used as references:

(i) https://doi.org/10.3390/rs14030559

(ii) https://doi.org/10.1016/j.sciaf.2023.e01577

(iii) https://doi.org/10.3390/s22197384

Response 3: The formulation of the research gap and review questions in the introduction is improved. We have carefully studied the three articles you recommended, and they have been referenced to enrich the research progress of this article.

Point 4: The conclusion is too long and needs to be rewritten to highlight the major findings of the review study in a concise manner (maximum of 200 words).

Response 4: Thank you for your advice. We have re-edited the conclusion of this article to ensure that the number of words is within 200 words. See the conclusion part of the article for details.

Reviewer 3 Report

Dear Authors,

Thanks for this review paper on CNN image classification and recognition in agriculture.

Please find here below suggestions and comments from review process:

General comment - this paper is submitted to Remote Sensing journal, however, majority of presented content is rather related to classification and recognition of imagery recorded closely - in laboratory conditions (seeds, post-harvest crops) of in fields by terrestrial machinery (crops, weeds detection). Still, it may be of interest of remote sensing community. I miss there information on availability of free software or free computational web applications. 

Abstract - it should clearly state to which time period this review refers.

Introduction - introduction to CNN is superficial (principles, constrains, position in image processing agriculture applications).

L66 - The the ....

Whole Crop Growth Cycle and CNN - not much information can be understood from Figure 1 without description of work principles. 

Application of CNN in the Whole Crop Growth Cycle - text should provide more specific information, in presented form it is too generic. 

L197 Please check grammar here: ... image, is an active research areas.

Discussion - Substantial part of discussion is independent from inforamtion provided in the text above. It should be reorganized.

L507 It sounds strange here: Compared with RGB images, MSI and HSI have less data, ... Please check and correct in line with information at L553.

L519-520 - The more commonly used satellites are the Sentinel-2 satellite of the United States.... Please correct information provided here - Sentinel-2 is  European satellite operated by European Space Agency (ESA).

Conclusions - please consider to indicate in tables on individual applications also information on "practical/experimnetal application". It is an important conclusion that most research outcomes have not been put to practical application and that robustness of models is low - information on this should be attributed to individual applications.

Author Response

Thank you very much for your valuable feedback on the manuscript, which will make it more comprehensive and rigorous. We have made revisions and improvements to the manuscript based on each of your proposed revisions. The details are as follows:

Point 1: General comment - this paper is submitted to Remote Sensing journal, however, majority of presented content is rather related to classification and recognition of imagery recorded closely - in laboratory conditions (seeds, post-harvest crops) of in fields by terrestrial machinery (crops, weeds detection). Still, it may be of interest of remote sensing community. I miss there information on availability of free software or free computational web applications.

Response 1: Thank you for your suggestion. It has been very helpful to us and we will do our best to improve the manuscript.

Point 2: Abstract - it should clearly state to which time period this review refers.

Response 2: We have added time period 2020-2023 to the abstract, as detailed in the abstract section.

Point 3: Introduction - introduction to CNN is superficial (principles, constrains, position in image processing agriculture applications).

Response 3: The principles of CNN were introduced in Figure 1, and the constraints and position in image processing agriculture applications of CNN were added in the Introduction. "CNN is the most popular framework among deep learning models applied in agricultural image processing, as it can quickly and accurately extract the most effective characterization from various features. One of the most significant constraints of CNN is that it relies on sufficient datasets with annotated labels, which need intensive datasets collection and manual labeling."

Point 4: L66 - The the ....

Response 4: We have removed the duplicate words in this sentence:"The application characteristics of CNN in different image-processing tasks are also analyzed and compared."

Point 5: Whole Crop Growth Cycle and CNN - not much information can be understood from Figure 1 without description of work principles.

Response 5: We have described the working principle of Figure 1, as detailed in the main text of the manuscript.

Point 6: Application of CNN in the Whole Crop Growth Cycle - text should provide more specific information, in presented form it is too generic.

Response 6: As per your suggestion, we have hanged the wording of the title: "Progress of CNN Applications in Crop Growth Cycle".

Point 7: L197 Please check grammar here: ... image, is an active research areas.

Response 7: We have modified the grammar of the sentence: "Crop plot classification based on the canopy with the use of remote sensing or UAV, is an active research area."

Point 8: Discussion - Substantial part of discussion is independent from inforamtion provided in the text above. It should be reorganized.

Response 8: Thank you very much for your valuable suggestion, but we think that the existing discussion has broken the division of crop growth stages. From another dimension, the image processing process, we have discussed and analyzed some common issues existing in the current research, which can enrich the analysis dimension of the article and provide more reference value. We hope to retain the existing discussion and gain your understanding.

Point 9: L507 It sounds strange here: Compared with RGB images, MSI and HSI have less data, ... Please check and correct in line with information at L553.

Response 9: We have corrected the information in this sentence: "Especially for MIS and HSI images with rich spectral information, and even simple models are difficult to over-fit."

Point 10: L519-520 - The more commonly used satellites are the Sentinel-2 satellite of the United States.... Please correct information provided here - Sentinel-2 is  European satellite operated by European Space Agency (ESA).

Response 10: We have corrected the information in this sentence: "The more commonly used satellites are the Sentinel-2 satellite of Europe and the GaoFeng (GF) series satellite of China."

Point 11: Conclusions - please consider to indicate in tables on individual applications also information on "practical/experimnetal application". It is an important conclusion that most research outcomes have not been put to practical application and that robustness of models is low - information on this should be attributed to individual applications.

Response 11: As per your suggestion, we have annotated the practical/experiential application in the table of the manuscript with a * sign. At the same time, we have removed inappropriate statements, as detailed in section 5 research prospect.

Reviewer 4 Report

First of all I’d like to highlight some of the features of this review paper.

This paper provides a thorough and systematic review of the use of convolutional neural networks (CNN) in various stages of the crop growth cycle, offering a holistic perspective of the field.

Effectively bridging the gap between machine learning and agriculture by demonstrating the application of advanced CNNs in modern agriculture.

Paper also analyzes the strengths and weaknesses of different CNNs, their suitability for specific stages, and how they can address specific problems.

Paper also discusses the application of CNNs in post-harvest stages, showcasing its potential in improving mechanized harvesting, product screening, and grading.

Paper also discusses the importance of crop growth cycle and relevant literature. Even thought this topic is rarely discussed.

The paper also can serve as a review of agriculture related datasets.

The overall paper is well structured and organized nicely and supported with proper citations.

Minor:

But a few things I’d suggest authors is to,

add information about public availability of the datasets as it is a major concern in development of precision agriculture systems.

also add some growth cycle and viewing angle related benchmark datasets in it discussion. For example, the following article could be appropriate choice for discussion as its recent benchmark dataset.

https://arxiv.org/abs/2305.10084 (CWD30 dataset)

I did not check it for plagiarism. Only minor checks are required.

Author Response

Thank you very much for your valuable feedback on the manuscript, which will make it more comprehensive and rigorous. We have made revisions and improvements to the manuscript based on each of your proposed revisions. The details are as follows:

Point 1: First of all I’d like to highlight some of the features of this review paper. This paper provides a thorough and systematic review of the use of convolutional neural networks (CNN) in various stages of the crop growth cycle, offering a holistic perspective of the field. Effectively bridging the gap between machine learning and agriculture by demonstrating the application of advanced CNNs in modern agriculture. Paper also analyzes the strengths and weaknesses of different CNNs, their suitability for specific stages, and how they can address specific problems. Paper also discusses the application of CNNs in post-harvest stages, showcasing its potential in improving mechanized harvesting, product screening, and grading. Paper also discusses the importance of crop growth cycle and relevant literature. Even thought this topic is rarely discussed. The paper also can serve as a review of agriculture related datasets. The overall paper is well structured and organized nicely and supported with proper citations.

Response 1: Thank you for your affirmation. We have also further optimized and adjusted the manuscript. Especially, academic comments have been added to the main part of the article, enriching existing research.

Point 2: add information about public availability of the datasets as it is a major concern in development of precision agriculture systems. also add some growth cycle and viewing angle related benchmark datasets in it discussion. For example, the following article could be appropriate choice for discussion as its recent benchmark dataset. https://arxiv.org/abs/2305.10084 (CWD30 dataset)

Response 2: Thank you for your valuable suggestion. We have added Table 5 to the manuscript to supplement the relevant datasets information currently collected. We discussed the CWD30 dataset and other datasets, and added them to the references.
